# α-galactosylceramide-stimulated invariant natural killer T-cells play a protective role in murine vulvovaginal candidiasis by *Candida albicans*

**Masahiro Abe**[1], **Yuki Kinjo**[2,3,4], **Sota Sadamoto**[1,5], **Minoru Shinozaki**[5], **Minoru Nagi**[1], **Kazutoshi Shibuya**[5], **Yoshitsugu Miyazaki**[1]*

1 Department of Chemotherapy and Mycoses, National Institute of Infectious Diseases, Tokyo, Japan,
2 Department of Bacteriology, The Jikei University School of Medicine, Tokyo, Japan, 3 Jikei Center for Biofilm Science and Technology, The Jikei University School of Medicine, Tokyo, Japan, 4 Department of Intelligent Network for Infection Control, Tohoku University Graduate School of Medicine, Sendai, Miyagi, Japan, 5 Department of Surgical Pathology, Toho University School of Medicine, Tokyo, Japan

* ym46@niid.go.jp

**Data Availability Statement:** All relevant data are within the paper and its Supporting Information files.

## Abstract

### Background

Vulvovaginal candidiasis is a common superficial candidiasis; however, a host's immunological mechanism against vaginal *Candida* infection remains unknown.

### Objectives

In this study, we aimed to elucidate the effect of iNKT cell activation on vulvovaginal candidiasis.

### Methods

Using a vulvovaginal candidiasis model with estrogenized mice, we evaluated the fungal burden and number of leukocyte infiltrations in the vaginal lavage of wild-type C57BL/6J mice after *Candida albicans* inoculation. One day before *C. albicans* inoculation, α-galactosylceramide (the α-GalCer group) or sterile phosphate-buffered saline (the sham group) was intraperitoneally injected into the mice. We also evaluated the level of antimicrobial peptide S100A8 in the vaginal lavage and analyzed the correlation between S100A8 concentration and the number of vaginal leukocyte infiltrations. Moreover, the number of uterine and vaginal immune cells were evaluated using flow cytometry.

### Results

The number of vaginal leukocyte infiltrations was significantly higher in the α-GalCer group than in the sham group 3 days after *C. albicans* inoculation. In addition, the fungal burden was significantly lower in the α-GalCer group than the sham group at 7 days after inoculation. In the analysis of S100A8 concentration of vaginal lavage, there were no significant differences between these two groups, although S100A8 concentration and the number of

**Funding:** Y.M. received funding from the Research Program on Emerging and Re-emerging Infectious Diseases of the Japan Agency for Medical Research and Development (AMED) Grant Number (JP21fk0108135, JP21fk0108139, and JP21fk0108094) https://www.amed.go.jp/en/index.html M.A. and Y.K. received funding from Japan Society for the Promotion of Science [KAKENHI] Grant Number 18K19529. https://www.jsps.go.jp/english/index.html M.A. received funding from Japan Society for the Promotion of Science [KAKENHI] Grant Number 20K17477. https://www.jsps.go.jp/english/index.html These funders had no role in study design, data collection and analysis, decision to publish, or preparation of the manuscript.

**Competing interests:** The authors have declared that no competing interests exist.

vaginal leukocyte infiltrations were positively correlated in the α-GalCer group. Moreover, the number of vaginal iNKT cells, NK cells and CD8$^+$ T-cells was significantly higher in the α-GalCer group 3 days after inoculation.

## Conclusions

α-GalCer-stimulated iNKT cells likely play a protective role against vulvovaginal candidiasis.

## Introduction

*Candida* species are opportunistic fungi that colonize the skin and mucocutaneous surfaces. Vulvovaginal candidiasis is a major type of superficial candidiasis that mainly occurs in child-bearing aged women [1, 2]. Under several conditions, including broad-spectrum antibiotic therapy, high estrogen levels, hyperglycemia, pregnancy, and immunosuppression, *Candida* species overgrow in the vagina and cause several symptoms, such as itchiness, redness, and discharge [1–4]. Vulvovaginal candidiasis affects approximately 75% of all women at least once during their lifetime; therefore, it is a common disease and strongly impacts their quality of life [1, 5, 6].

*Candida albicans* is the most common cause of vulvovaginal candidiasis and causes the most noticeable symptoms. Although the isolation rate of *C. albicans* differs among investigations. Various epidemiological studies identified *C. albicans* in 60%–90% of the vaginal *Candida* isolates [1, 2, 7–9]. In addition, studies in the mouse vulvovaginal candidiasis model showed that the vaginal inoculation of *C. albicans* led to the most neutrophil recruitment and mucosal damage, which were thought to be linked to clinical symptoms [10]. The recruited neutrophils are reported to be the major effector cells in vulvovaginal candidiasis; however, the function of the neutrophils against vaginal *Candida* species remains unclear [11, 12]. In addition, while several factors have been reported to associate with vaginal immunity against vulvovaginal candidiasis, there is still much ambiguity in this field [13–15].

Invariant natural killer T-cells (iNKT cells) are innate-type lymphocytes expressing an invariant T-cell receptor (TCR)-α chain; they are involved in both innate and adaptive immunity by secreting several kinds of cytokines in response to CD1d-presented glycolipids, such as α-galactosylceramide (α-GalCer) [16]. Because of their cytokine-producing profile, iNKT cells were reported to play important roles against several microbial infections [17–20]. For example, previous studies reported the immunomodulatory functions of iNKT cells against *Candida* infection in the systemic dissemination mouse model [21, 22]. In the female reproductive system, iNKT cells were reported to play several roles in the uterus, vagina, and decidua [23–28]. Collectively, iNKT cells are thought to be strongly associated with the diseases of the female reproductive system. However, there are no studies describing the association between *C. albicans* vulvovaginal candidiasis and iNKT cells.

In this report, we investigated the function of α-GalCer-stimulated iNKT cells against *C. albicans* vaginal infections. We used the estrogen-dependent pseudoestrus vulvovaginal candidiasis model to evaluate the vaginal fungal burden by *C. albicans*, the accumulations of leukocytes, and the secretion of an antimicrobial peptide. We also evaluated accumulated uterine and vaginal immune cells after the vaginal inoculation of *C. albicans* using flow cytometry.

## Material and methods

### Mice

Female, 6–7-week-old C57BL/6J mice (purchased from Japan SLC, Inc., Shizuoka, Japan) were maintained under specific pathogen-free conditions at the National Institute of Infectious

Diseases in Japan. All the experiments were reviewed and approved by the Animal Care and Use Committee of the National Institute of Infectious Diseases. All of the mice were euthanized by the inhalation of carbon dioxide after experiments. Experimental protocols were designed to minimize animal suffering and limit the number of animals used in an experiment (Approval number: 120051).

## Yeast strains and preparation

*Candida albicans* strain SC5314 was used as the reference strain for the mouse vulvovaginal candidiasis model. *C. albicans* was grown at 30˚C on yeast extract peptone dextrose (YPD) agar. Then *C. albicans* was grown at 30˚C in YPD broth for 18–24 h. Afterward, the yeast cells were collected, washed, and resuspended in sterile phosphate-buffered saline (PBS) at approximately $5.0 \times 10^8$ colony-forming units (CFU)/mL.

## Mouse vulvovaginal candidiasis model

The established mouse vulvovaginal candidiasis model was used with minor modifications [10, 11, 13]. Briefly, 0.2 mg of β-estradiol 17-valerate (Nacalai Tesque, Kyoto, Japan) dissolved in 0.1 mL of sesame oil was subcutaneously administrated to each mouse 1 week before *C. albicans* inoculation. The administration of β-estradiol 17-valerate was repeated once a week until the end of the experiments. The estrogenized mice were then intravaginally inoculated with 20 μL, at approximately $1.0 \times 10^7$ CFU/mouse, of prepared *C. albicans* suspension to induce vulvovaginal candidiasis.

## Reagent preparation and iNKT cells stimulation

α-GalCer (KRN7000; Funakoshi Co., Tokyo, Japan) was dissolved at 1 mg/mL in the vehicle solution composed of a buffer (pH 7.2), containing 57 mg/mL sucrose, 7.5 mg/mL histidine, and 5 mg/mL Tween 20. α-GalCer was diluted with sterile PBS to 5 μg/mL before injection. One day before *C. albicans* inoculation, 200 μL of 5 μg/mL α-GalCer was intraperitoneally injected into the mice (the α-GalCer group). On the other hand, the vehicle solution was diluted with sterile PBS, similarly to the α-GalCer solution. Then, 200 μL of the diluted vehicle was intraperitoneally injected into the mice (the sham group).

## Evaluation of vaginal fungal burden and infiltration of leukocytes

Vaginal lavage was collected from a mouse under isoflurane anesthesia by gently washing the vagina twice with 50 μL of sterile PBS per wash. The aspirated vaginal lavage fluid was serially diluted, and 50 μL of each dilution was plated on YPD agar with penicillin/streptomycin. These plates were incubated for 24 h at 30˚C before counting the fungal colonies. The vaginal leukocyte infiltrations were stained with trypan blue and counted under a light microscope using a hemocytometer. In Papanicoloau staining, 10 μL of vaginal lavage was spotted on slide glass, fixed with M-FIX® spray (Merck, Darmstadt, Germany), and stained to evaluate leukocyte infiltrations. The infiltrated leukocytes were counted in at least 5 randomly selected fields using microscopy with high power field (40x objective). The remaining vaginal lavage was centrifuged and stored at −30˚C for cytokine analysis.

## Evaluation of S100A8 concentration in vaginal lavage

The cytokines in the vaginal lavage were measured using commercial S100A8 enzyme-linked immunosorbent assay kits (R&D Systems, Minneapolis, MN, United States) according to the manufacturer's protocol.

## Immune cells isolated from vagina or uterus

After the *C. albicans*-inoculated mice were euthanized, their uteruses and vaginas were carefully isolated, separated, and incubated with digestive enzymes as previously described with minor modifications [29, 30]. In short, the vagina was incubated with 0.5 mg/mL Dispase II (Roche Applied Science, Penzberg, Germany) for 15 min at 37˚C, and then minced and incubated with 0.425 mg/mL Collagenase D (Roche Applied Science, Penzberg, Germany) and 30 μg/mL DNase I (Sigma-Aldrich, St. Louis, MO, United States) at 37˚C for 60 min with rotation. The uterus was minced and incubated with 0.25 mg/mL Collagenase D and 2.5 μg/mL DNase I at 37˚C for 45 min with rotation. After enzyme digestion, EDTA at the final concentration of 5mM was added to the tissues to incubate for 5 min to stop the digestion. These digestive enzyme-treated tissues were then gently homogenized and passed through a 70-μm nylon-mesh filter to obtain single immune cells.

## Flow cytometry analysis

The vaginal and uterine inflammatory cells were blocked with purified an anti-mouse CD16/32 antibody (clone 93; Biolegend), an Fc-receptor blocking antibody, and stained with various fluorochrome-conjugated secondary antibodies (Biolegend; listed in Table 1) or mouse CD1d tetramer-PE (Medical & Biological Laboratories). The immune cells were enumerated using the FACS Canto IIⓇ Cell Analyzer (BD Biosciences). These data were analyzed using FlowJo, version 10.6.1 (Tree Star, Inc, Ashland, OR, USA).

## Statistical analysis

The continuous variables of the two groups were compared with equal variance using the Student *t*-test. If the standard deviations differed between the two groups, Mann Whitney *U*-test was used. The correlations were analyzed with Pearson's test. A *P*-value of less than 0.05 from the two-tailed test was considered significant for all the tests. Statistical analyses were performed using GraphPad Prism, version 8 (GraphPad Software, La Jolla, CA, United States).

**Table 1. The antibodies used in this study.**

| Antigen | Clone | Conjugated fluorochrome |
|---|---|---|
| CD45 | 30-F11 | FITC |
| CD11c | N418 | PE |
| NK1.1 | PK136 | PE |
| Ly6C | HK1.4 | PE/Cy7 |
| CD3ε | 145-2C11 | PE/Cy7 |
| F4/80 | BM8 | APC |
| TCRγδ | GL3 | APC |
| I-A/I-E | M5/114.15.2 | AF700 |
| CD4 | GK1.5 | AF700 |
| TCRβ | H57-597 | AF700 |
| CD11b | M1/70 | BV421 |
| CD8α | 53–6.7 | BV510 |
| Ly6G | 1A8 | BV510 |
| CD19 | 6D5 | BV510 |

FITC: Fluorescein isothiocyanate; PE: Phycoerythrin; PE/Cy7: Phycoerythrin-cyanin7; APC: Allophycocyanin; AF700: Alexa Fluor 700; BV421: Brilliant Violet 421; BV510: Brilliant Violet 510.

## Results

### Earlier clearance of vaginal *C. albicans* burden by α-GalCer stimulation

We evaluated the vaginal fungal burden of *C. albicans* by intravaginally inoculating each mouse with $1 \times 10^7$ of *C. albicans* cells and comparing the number of fungal cells in the α-GalCer and the sham groups. At 3 days after inoculation, the fungal burden was not significantly different between the two groups (Fig 1A). Over time, the fungal burden in both groups gradually decreased; at 7 days after inoculation, the number of vaginal *C. albicans* in the α-GalCer group became significantly lower than in the sham group (Fig 1B). At 14 days after inoculation, the mean fungal burden was still lower in the α-GalCer group; there was no significant difference between the groups (Fig 1C). These data indicated that α-GalCer stimulation would lead to the earlier clearance of vaginal *C. albicans*, likely due to iNKT cell activation.

### Earlier infiltration of leukocytes into the vagina by α-GalCer stimulation

We also evaluated the infiltration of the vaginal leukocytes by counting the number of vaginal leukocyte infiltrations using trypan blue staining and a hemocytometer. The number of leukocyte infiltrations in vaginal lavage of the α-GalCer group was significantly higher than that in the sham group 3 days after inoculation (Fig 2A). On the other hand, the differences between the two groups became smaller over time; the number of leukocyte infiltrations in the vaginal lavage was not significantly different between the groups at 7 and 14 days after inoculation (Fig 2B and 2C). Additionally, we also evaluated vaginal leukocyte infiltrations using Papanicoloau staining 3 days after inoculation, based on the above results. The number of leukocyte infiltrations was significantly higher in the α-GalCer group than that in the sham group, which was accordant with the results of Trypan blue staining (Fig 2D). Moreover, the correlation analysis showed the number of leukocyte infiltrations had significantly positive correlations between Papanicolaou staining and Trypan blue staining, not only in the α-GalCer group but also in the sham group (Fig 2E). Collectively, these data suggested that iNKT cell activation by α-GalCer would result in the earlier infiltration of leukocytes into the vagina.

### Elevated level of antimicrobial peptide S100A8 in vaginal lavage was correlated with leukocyte infiltrations after α-GalCer injection

Next, we examined if *C. albicans* inoculation stimulated the release of antimicrobial peptides. We evaluated the concentration of antimicrobial peptide S100A8 in the vaginal lavage at 3 and

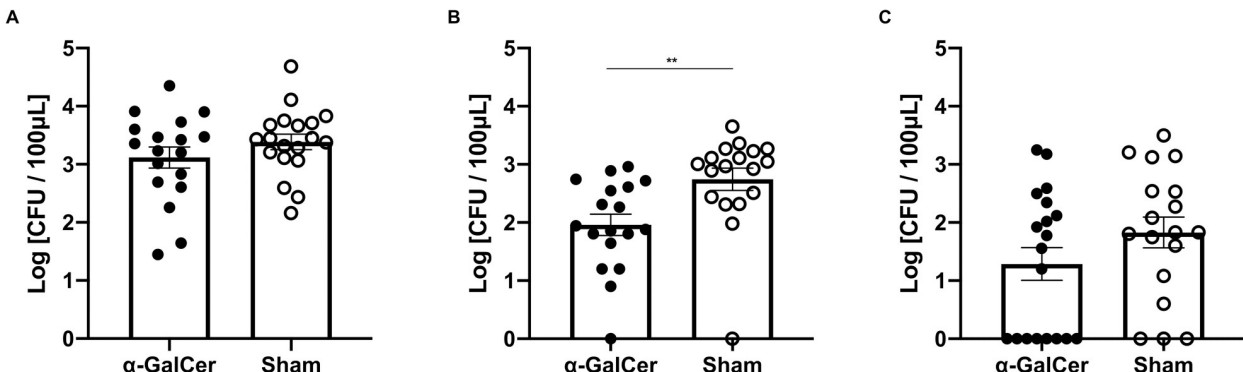

**Fig 1. Vaginal fungal burden was significantly lower in the α-GalCer group 7 days after *C. albicans* inoculation.** (A–C) The burden of *C. albicans* in the vaginal lavage 3 (A), 7 (B), and 14 days (C) after inoculation. Fungal burdens in the vaginal lavage were shown as Log [CFU/100μL]. All the results were from at least three independent experiments, with a total of 18–19 samples per group, and were expressed as mean ± standard error of the mean. $^{**}P < 0.01$.

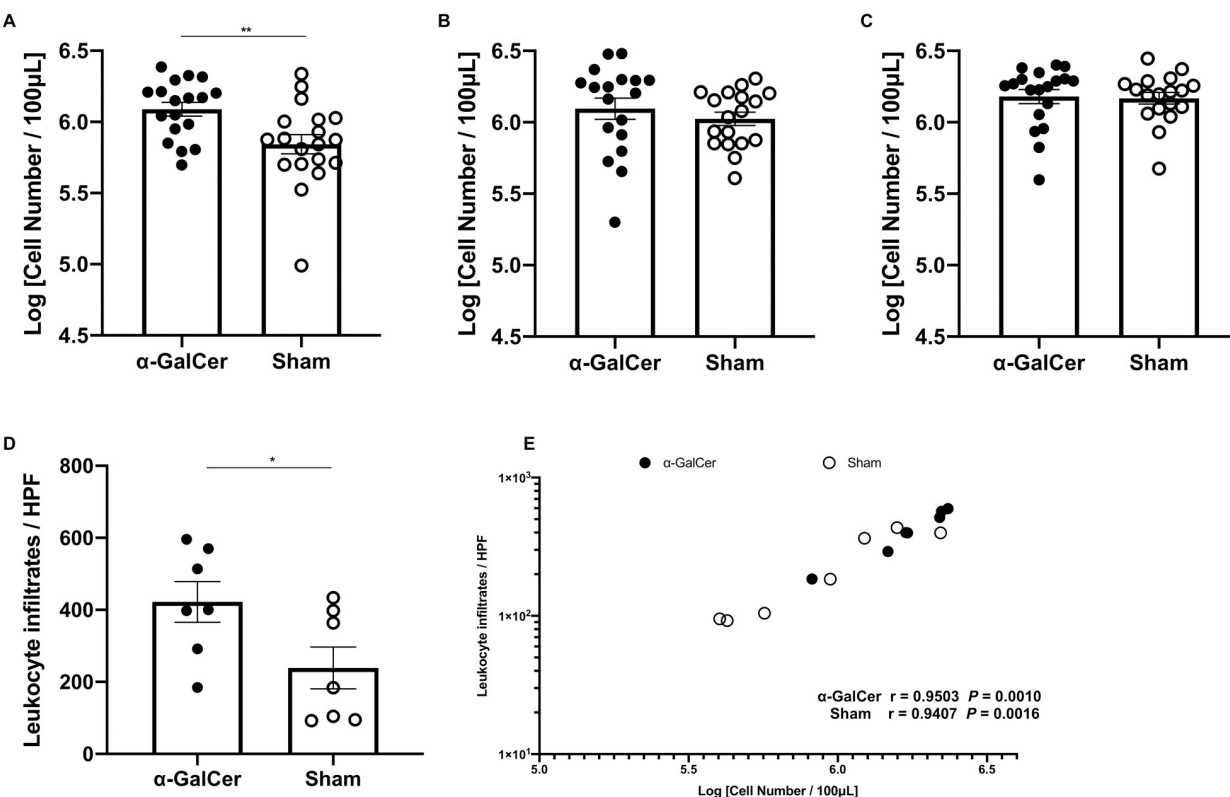

**Fig 2. The number of vaginal leukocyte infiltrations was significantly higher in the α-GalCer group 3 days after *C. albicans* inoculation.** (A −E) The number of leukocyte infiltrations counted by Trypan blue staining in the vaginal lavage 3 (A), 7 (B), and 14 days (C) after inoculation. The number of leukocyte infiltrations in the vaginal lavage was shown as Log [Cell Number/100μL]. These results were from at least three independent experiments, with a total of 18–19 samples per group, and were expressed as mean ± standard error of the mean. The number of leukocyte infiltrations counted by Papanicolaou stain (D) and correlations between Papanicolaou staining and Trypan blue staining (E). These results were from two independent experiments, with a total of 7 samples per group, and the number of leukocyte infiltrations / High power field (HPF) was expressed as mean ± standard error of the mean. "r" denotes Pearson's correlation coefficient. $^{*}P < 0.05$, $^{**}P < 0.01$.

7 days after *C. albicans* inoculation. The means S100A8 concentration on both days were higher in the α-GalCer group, although there were no significant differences between the two groups on either day (Fig 3A and 3B). Then, we investigated the correlation between the number of vaginal leukocyte infiltrations and S100A8 concentration in the vaginal lavage. There were no correlations between the two factors in the sham group 3 or 7 days after inoculation (Fig 3C and 3D). On the other hand, there was a significant positive correlation between the number of vaginal leukocyte infiltrations and S100A8 concentration in the α-GalCer group at both time points (Fig 3C and 3D). These results suggested that the antimicrobial peptide S100A8 was elevated in proportion to the increase in the number of infiltrating vaginal leukocyte infiltrations, especially after α-GalCer injection; these two factors might synergistically cause the earlier clearance of vaginal *C. albicans*.

## Vaginal NK cells and CD8⁺ T-cells increase after α-GalCer injection

In addition to studying leukocyte infiltrations in the vaginal lavage, we evaluated the immune cells in the vagina and uterus using flow cytometry. Based on the observed number of leukocyte infiltrations in the vaginal lavage, we evaluated vaginal and uterine immune cells 3 days after *C. albicans* inoculation. There were no significant differences in the number of uterine immune cells between the two groups (S1 Fig). In contrast, the number of vaginal NK cells

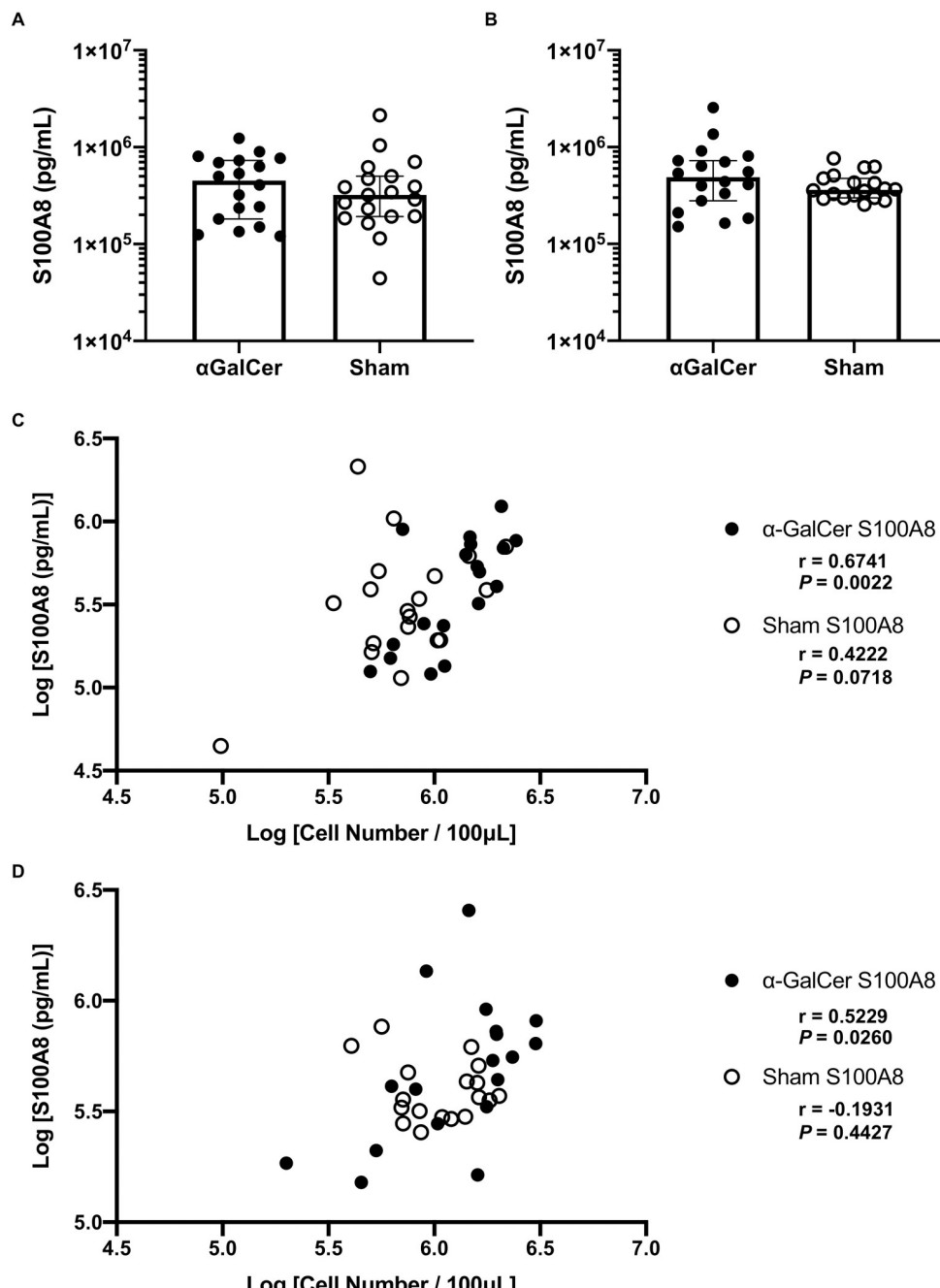

**Fig 3. Significant positive relationships between S100A8 concentration and the number of vaginal leukocyte infiltrations in the α-GalCer group.** (A–D) Vaginal lavage concentration of S100A8 3 (A) and 7 days (B) after vaginal inoculation of *C. albicans*, and the relationships between S100A8 concentration (pg/mL) and the number of leukocyte infiltrations in vaginal lavage 3 (C) and 7 days (D) after the vaginal inoculation of *C. albicans*. S100A8 concentration is shown as Log [pg/mL] and the number of leukocyte infiltrations as Log [Cell Number/100μL] in the correlation plot between S100A8 concentrations and leukocyte infiltration numbers. The results of at least three independent experiments, each with 18–19 samples, were pooled for analysis. S100A8 concentration is expressed as median ± 95% confidential interval. "r" denotes Pearson's correlation coefficient.

(CD11b⁻, NK1.1⁺, and CD3ε⁻ cells) and CD8⁺ T-cells (CD11b⁻, CD3ε⁺, and CD8α⁺ cells) were significantly increased in the α-GalCer-treated group compared to those in the sham group (Fig 4A). The number of vaginal γδT-cells (CD11b⁻, CD3ε⁺, CD8α⁻, CD4⁻, and

TCRγδ⁺ cells) tended to be higher in the α-GalCer group than in the sham group, although there was no significant difference. Regardless of the leukocyte infiltrations in vaginal lavage, there were no differences in the number of vaginal neutrophils (CD11b⁺ and Ly6G⁺ cells) and other immune cells between the two groups. In addition, the number of iNKT cells (CD11b⁻ ˜ int, CD1d tetramer⁺, CD19⁻ and TCRβ⁺ cells) and percentage of iNKT cells to CD45⁺ cells was significantly higher in the α-GalCer group than that of the sham group (Fig 4B and 4C). These results implied that α-GalCer injection resulted in the increase of vaginal iNKT cells, NK cells

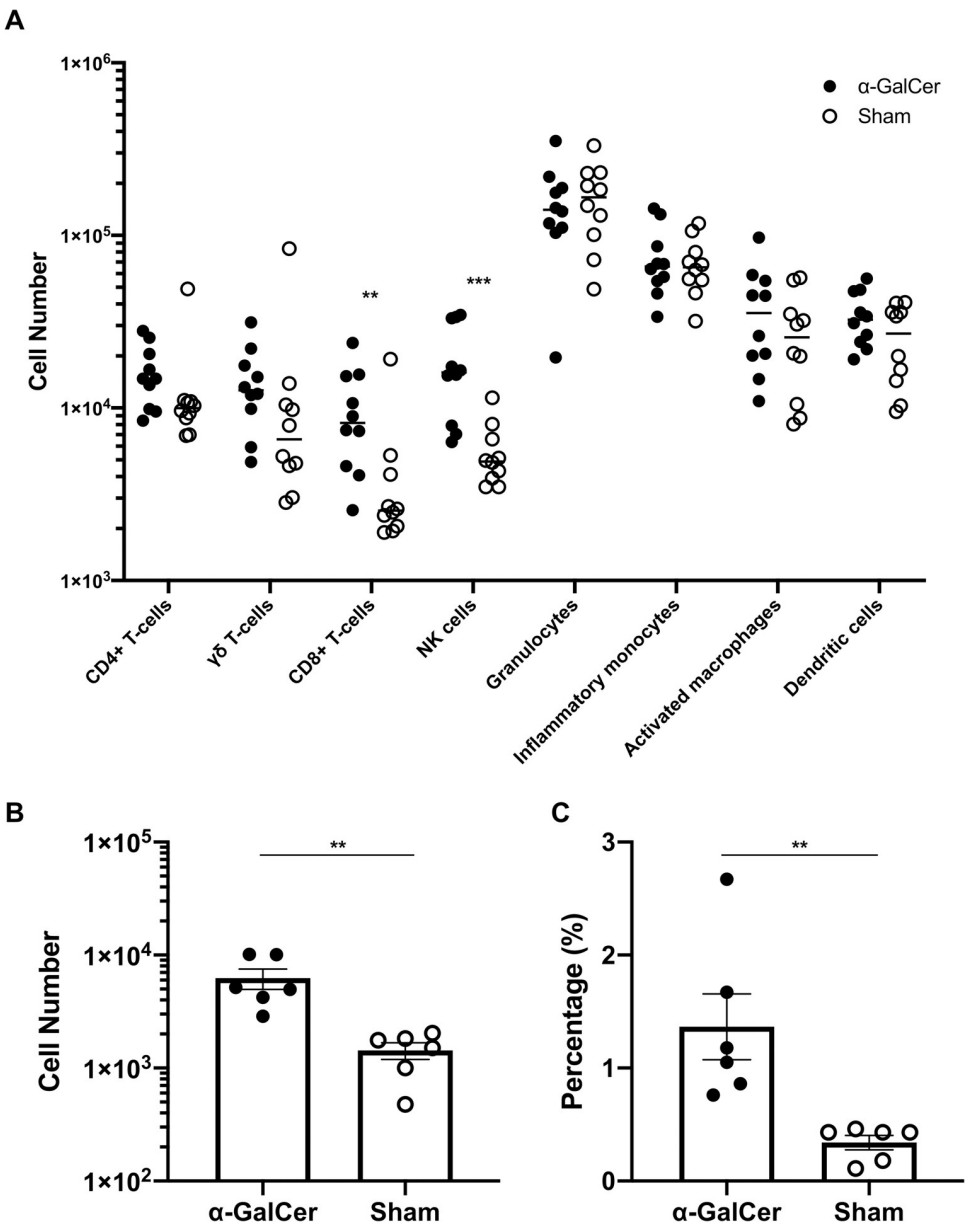

**Fig 4. The numbers of vaginal iNKT, NK and CD8+ T-cells were significantly higher in the α-GalCer group.** (A-C) The number of vaginal (A) immune cells 3 days after *C. albicans* inoculation. The results of three independent experiments, each with 10 samples, were pooled for analysis. The number (B) and percentage per CD45⁺ cells (C) of vaginal iNKT cells 3 days after *C. albicans* inoculation. The results of two independent experiments, each with 6 samples, were pooled for analysis. ** $P < 0.01$, *** $P < 0.001$.

and CD8[+] T-cells, and these cells might be cooperatively related with the defense mechanism against vaginal *C. albicans*.

## Discussion

Vulvovaginal candidiasis is one of the common mucocutaneous candidiasis in humans, and *Candida albicans* is reported to be the most common causative pathogen [1–4, 7–9]. In the vulvovaginal candidiasis mouse model, *C. albicans* could strongly induce neutrophil recruitment; however, the function of these neutrophils and other factors associated with vulvovaginal candidiasis remains unclear [11–15]. Moreover, although iNKT cells are associated with immunity against several genitourinary diseases, their role in vulvovaginal candidiasis is unknown. In our study, α-GalCer stimulation before vaginal *C. albicans* inoculation resulted in the earlier recruitment of leukocytes and the earlier clearance of vaginal *C. albicans*. The level of antimicrobial peptide S100A8 in vaginal lavage was elevated concomitantly with the number of vaginal leukocyte infiltrations. Additionally, the number of vaginal iNKT cells, NK cells and CD8[+] T-cells increased in α-GalCer-treated mice, suggesting that the protective role of iNKT cells against *C. albicans* vaginal infections is linked to antimicrobial peptide secretion and the increase of NK cells and CD8[+] T cells. To our best knowledge, this is the first report describing the protective role of iNKT cells in vulvovaginal candidiasis.

The analysis of vaginal leukocyte infiltrations and fungal burden revealed that the recruitment of leukocytes occurred earlier in the α-GalCer group than the sham group. It is known that iNKT cells promptly secret many cytokines after α-GalCer stimulation, resulting in the earlier accumulation of leukocytes. The fungal burden in the vagina was significantly lower in the α-GalCer-treated group at 7 days after *C. albicans* inoculation, likely due to the earlier recruitment of leukocytes. However, there were no significant differences in the number of vaginal leukocyte infiltrations between the two groups at 7 and 14 days after *C. albicans* inoculation. It was assumed that *C. albicans* recruit neutrophils robustly after vaginal infection, resulting in the accumulated of many leukocytes in the vagina not only in the α-GalCer group but also in the sham group over time [10, 11]. In addition, the difference in the vaginal leukocyte infiltrations between the α-GalCer group and the sham group 3 days after inoculation was relatively small. It was presumed that the cytokine secretion by iNKT cells after α-GalCer stimulation is likely quite rapid; therefore, the recruitment of leukocytes into the vagina might occur earlier than 3 days after inoculation. Further investigation is necessary to elucidate this point.

In our study, α-GalCer stimulation resulted in improvement of vulvovaginal candidiasis, although several previous reports about roles of iNKT cells against infections showed that α-GalCer stimulation resulted in exacerbation of infections [21, 22]. The reason for these contradictory outcomes might be due to the difference in the locus of infection; systemic infection or mucosal infection. The previous reports about exacerbated outcomes focused on systemic infections, and α-GalCer-stimulated iNKT cells played an immunosuppressive role in these kinds of infections [21, 22]. On the other hands, α-GalCer-stimulated iNKT cells were also reported to play a protective role in mucosal or localized infections [17, 23]. Although a role of iNKT cells depend on types of infection, α-GalCer-stimulated iNKT cells was assumed to play a protective role in our vulvovaginal candidiasis by *C. albicans*.

The analysis of antimicrobial peptide S100A8 in the vaginal lavage showed that there were no significant differences between the two groups at 3 or 7 days after inoculation. Because *C. albicans* is pathogenic and capable of inducing inflammation robustly, many antimicrobial peptides may be secreted not only in the α-GalCer group but also in the sham group. However, the correlation analysis showed a significant positive correlation in the α-GalCer group at 3

and 7 days after inoculation, but no significant correlations in the sham group. Previous reports have shown that S100A8 is associated with many infectious diseases and strongly induces neutrophils recruitment [31, 32]. In vulvovaginal candidiasis, it was reported that infiltrating cells into vaginal lumen after *C. albicans* inoculation were predominantly neutrophils, and these cells were main sources of S100A8; however, these factors had no apparent effect on vaginal fungal burden [33, 34]. In our study, S100A8 increased correlated with leukocyte infiltrations in the early phase of infection, especially in α-GalCer-stimulated mice; although exact mechanisms against vaginal *C. albicans* infection remain unknown. In a previous study, S100A8 was reported to be the major antifungal component in neutrophil extracellular traps (NETs); its absence in NETs resulted in the complete loss of antifungal activity [35]; however, further study is necessary to clarify an exact protective role of α-GalCer-stimulated iNKT cells against our vulvovaginal candidiasis model.

Our flow cytometry analysis data demonstrated no significant differences in the number of immune cells between the α-GalCer group and the sham groups in the uterus. In a previous study, uterine γδT-cells played protective roles in vulvovaginal candidiasis by *C. albicans*, although our results showed that uterine γδT-cells did not differ between the α-GalCer group and the sham groups [36]. On the other hands, the number of vaginal iNKT cells significantly increased in the α-GalCer-stimulated mice. In addition, the number of vaginal NK cells and CD8+ T-cells was also increased in the α-GalCer group. The iNKT cells secret a large amount of IFN-γ upon α-GalCer stimulation, and NK cells and CD8+ T-cells are known to be activated by IFN-γ stimulation [37–39]. In addition, NK cells and CD8+ T-cells also play crucial roles in orchestrating inflammation and defense mechanisms against infection by *Candida* species [40–43]. Collectively, the increase of vaginal iNKT cells after α-GalCer stimulation and subsequent increase of NK cells and CD8+ T-cells was assumued to play important roles against vaginal *C. albicans* infections.

There are several limitations to our study. First, we only used one reference strain of *C. albicans*; therefore, the involvement of iNKT cells in vulvovaginal candidiasis caused by non-albicans *Candida* or other strains of *C. albicans* remains unknown. However, *C. albicans* is the most common causative species of vulvovaginal candidiasis; hence, the analysis of *C. albicans* is likely most important and clinically relevant. Second, we only analyzed the results at 3, 7, and 14 days after *C. albicans* inoculation; we could not analyze the changes over time, especially the infiltration of leukocytes between 0 and 3 days after inoculation. In addition, the vaginal fungal burden seemed to be lower than that in the previous reports [10, 11, 34]. However, *C. albicans* was detected from vaginal lavage 3-days after inoculation, which meant no mice had lack of colonization in our study. The differences of fungal burden might be affected by the differences of the environments of the facilities, and vaginal microflora. In addition, previous reports showed that the genetic differences of susceptibilities to vulvovaginal candidiasis depended on the murine strain [44, 45]. In human, symptoms of vulvovaginal candidiasis vary considerably from person to person, therefore, our models are rather close to clinical settings and might be useful. Third, our study does not provide direct evidence that iNKT cells ameliorate vulvovaginal candidiasis. It was presumed that cytokine secretion by α-GalCer-stimulated iNKT cells increased the number of vaginal NK cells and CD8+ T cells, accumulating leukocyte infiltrations and increased S100A8, which resulted in improvement of vulvovaginal candidiasis. It remains unknown whether α-GalCer-stimulated iNKT cells, or NK cells/CD8+ T-cells affect the expression of S100A8. Our results implicated that leukocyte infiltrations and S100A8 expression accordant with iNKT cells, NK cells, and CD8+ T-cells played coordinately protective roles against vulvovaginal candidiasis, although further investigation is necessary to elucidate these points.

In conclusion, our results indicate that the stimulation of iNKT cells by α-GalCer injection leads to the earlier leukocyte infiltrations, the elevation of antimicrobial peptide S100A8, and the increase in the number of vaginal iNKT cells, NK cells and CD8[+] T cells, resulting in the earlier clearance of vaginal *C. albicans*. Our investigation suggested a strategy of controlling vulvovaginal candidiasis by targeting iNKT cells.

## Supporting information

**S1 Fig. The numbers of uterine immune cells did not differ between the α-GalCer group and sham group.** The number of uterine immune cells 3 days after *C. albicans* inoculation. The results of three independent experiments, each with 10 samples, were pooled for analysis. (TIF)

## Author Contributions

**Conceptualization:** Masahiro Abe.

**Data curation:** Masahiro Abe.

**Formal analysis:** Masahiro Abe.

**Funding acquisition:** Masahiro Abe, Yuki Kinjo, Yoshitsugu Miyazaki.

**Investigation:** Masahiro Abe, Sota Sadamoto, Minoru Shinozaki.

**Methodology:** Masahiro Abe, Yuki Kinjo, Yoshitsugu Miyazaki.

**Project administration:** Yoshitsugu Miyazaki.

**Resources:** Yuki Kinjo, Kazutoshi Shibuya, Yoshitsugu Miyazaki.

**Supervision:** Yoshitsugu Miyazaki.

**Visualization:** Masahiro Abe.

**Writing – original draft:** Masahiro Abe.

**Writing – review & editing:** Yuki Kinjo, Sota Sadamoto, Minoru Shinozaki, Minoru Nagi, Kazutoshi Shibuya, Yoshitsugu Miyazaki.

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
