## [Decision Letter · Decision Letter 0]

3 Aug 2021

PONE-D-21-16033

Invariant natural killer T-cells play a protective role in murine vulvovaginal candidiasis by *Candida albicans*

PLOS ONE

Dear Dr. Miyazaki,

Thank you for submitting your manuscript to PLOS ONE. After careful consideration, we feel that it has merit but does not fully meet PLOS ONE’s publication criteria as it currently stands. Therefore, we invite you to submit a revised version of the manuscript that addresses the points raised during the review process.

We look forward to receiving your revised manuscript.

Kind regards,

Karen L. Wozniak, PhD

Academic Editor

PLOS ONE

2. Please check to ensure they have included a completed ARRIVE checklist as a Supporting Information file.

3. To comply with PLOS ONE submissions requirements, in your Methods section, please provide additional information regarding the experiments involving animals and ensure you have included details on (1) methods of sacrifice, (2) methods of anesthesia and/or analgesia, and (3) efforts to alleviate suffering.

(Conflict of Interest Statement)

“This work was supported in part by the Research Program on Emerging and Re-emerging Infectious Diseases of the Japan Agency for Medical Research and Development (AMED) under Grant Number (JP20fk0108135, JP20fk0108094), and by JSPS KAKENHI Grant Number 20K17477 and 18K19529.”

“Y.M. received funding from the Research Program on Emerging and Re-emerging Infectious Diseases of the Japan Agency for Medical Research and Development (AMED)

Grant Number (JP20fk0108135, and JP20fk0108094)

https://www.amed.go.jp/en/index.html

M.A. and Y.K. received funding from Japan Society for the Promotion of Science [KAKENHI] Grant Number 18K19529.

https://www.jsps.go.jp/english/index.html

M.A. received funding from Japan Society for the Promotion of Science [KAKENHI] Grant Number 20K17477.

https://www.jsps.go.jp/english/index.html

These funders had no role in study design, data collection and analysis, decision to publish, or preparation of the manuscript.”

Additional Editor Comments (if provided):

Reviewers' comments:

Reviewer's Responses to Questions

**Comments to the Author**

1. Is the manuscript technically sound, and do the data support the conclusions?

Reviewer #1: Partly

Reviewer #2: Partly

2. Has the statistical analysis been performed appropriately and rigorously? 

Reviewer #1: Yes

Reviewer #2: Yes

3. Have the authors made all data underlying the findings in their manuscript fully available?

Reviewer #1: Yes

Reviewer #2: Yes

4. Is the manuscript presented in an intelligible fashion and written in standard English?

Reviewer #1: Yes

Reviewer #2: No

5. Review Comments to the Author

Reviewer #1: In this manuscript, the authors investigated a role of iNKT cells as a mediator of immune responses that could contribute to fungal clearance in a mouse model of C. albicans vaginal infection. Samples of vaginal lavage fluid were collected from estrogenized mice pre-treated with α-GalCer one day prior to vaginal inoculation and examined for fungal loads, cellular infiltrates and levels of S100A8. Leukocytes isolated from uterine and vaginal tissues were analyzed for respective cell-surface markers for identification. Results from lavage fluid indicated a reduction in vaginal fungal burden on day 7 p.i. and an increase in leukocyte migration on day 3 p.i. with a positive correlation between the presence of S100A8 and leukocyte counts. An analysis of uterine- or vaginal-associated cells by flow cytometry showed elevated numbers of CD8+ T cells and NK cells in the vagina from mice treated with α-GalCer while leukocyte profiles in the uterus were similar between the sham and α-GalCer groups. Collectively, the authors concluded that iNKT cells, presumably activated by the α-GalCer treatment, mediated a protective inflammatory response by induction of S100A8 through a mechanism involving CD8+ T cells and NK cells that ultimately resulted in reduction in vaginal C. albicans burden. Despite the rigorous flow cytometry analyses, there are several weaknesses and missing elements in the methodology, and overall, the study appears incomplete to support the conclusions. The following are major issues found in the manuscript.

1. The cited references in regard to priming of iNKT cells by α-GalCer demonstrate immunosuppressive properties. As shown in (21), activation of iNKT cells by intraperitoneal α-GalCer treatment resulted in reduced phagocytic capacity and increased C. albicans tissue invasion/deaths while CD1d KO mice (NKT cells-deficient) showed higher fungal clearance/survival. α-GalCer treatment alone was indicated to have a neutropenic effect (22). The contradictory outcomes in the vaginal immune response should be addressed.

2. As mentioned by the authors, the effect of α-GalCer stimulation on iNKT cells appears to occur as fast as 2-3 h post i.p. injection. The literature also suggests that iNKT activation is short living. A kinetic study showed that iNKT activation, measured by CD69 expression as a marker, peaked at 12 h post injection and returned to the baseline level within 72 h (21). Similar results were seen in fungal (22) and HSV2 (23) loads increased within 2 days. A study in (24) included α-GalCer injection on -2, 0, 3, 7 days post-infection and showed no difference in chlamydia burden. Thus, the effect of α-GalCer on iNKT cell activation on the time points is likely minimal and should be optimized/monitored/confirmed in the mouse VVC model.

3. Despite the high estrogen dose and C. albicans inoculum, seemingly higher ends of the standard range, it is concerning that animals failed sustain vaginal fungal burden for a period of 14 days in conventional C57BL6 mice. Although SC5314 strain is not a robust colonizer of the vaginal mucosa, estrogenized mice should maintain consistent levels of colonization for 14 days, and longer in most cases. Or the data are not interpretable if there is no distinction between clearance and lack of colonization.

4. It is unclear whether trypan blue dye was used for the purpose of cell viability staining or pan-nuclear staining. Since trypan blue dye is only permeable in cells the lacking intact membrane integrity (i.e. dead), it is only helpful in identifying dead cells. To accurately quantify a cell infiltrate, a staining method that aids visualization of the nuclear morphology (e.g. H&E or Pap smear) should be used. On the same note, “inflammatory cells” should be reworded to “leukocyte infiltrates”.

5. Previous studies showed that infiltrating cells into the vaginal lumen following vaginal C. albicans inoculation are predominantly neutrophils [PubMed 15102806], are the main source of S100A8 detected in vaginal secretions (11) and have no apparent effect on fungal burden in mice [28292981]. This information should be addressed in Discussion. Furthermore, it is unknow whether iNKT cell priming by α-GalCer or any downstream effectors (CD8+ T cells and NK cells) have impact on the expression of S100A8. Since S100A8 is the sole parameter of antifungal activity in the current study, this should be confirmed experimentally.

6. What is the rationale for evaluating the immune cells in the uterus in conjunction with the vaginal cells? C. albicans from the vaginal origin rarely invades the upper reproductive tract or does not lead to infection. If so, this should be reflected in the data from the uterine immune cells. If not, the data are not relevant in the current study and should be removed.

Reviewer #2: In the manuscript, Abe et al. studied the host defense with a focus on the effect of iNKT cell activation in a murine model of vulvovaginal candidiasis. They found that mice receiving α-GalCer, an iNKT cell agonist, control fungal pathogen better than mice receiving PBS. This better fungal clearance is accompanied by increased CD8+ T cells and NK cells. There is also a trend in increased S100A8 production but it does not reach statistical significance. The overall research design and results are solid. There are several concerns about this manuscript:

A major concern is that it is unclear whether iNKT cells are induced or activated in their model, for example, if their cell number changes, or if they produce more cytokines after stimulation.

Another major concern is that the author presents the data without explaining clearly the purpose or the meanings. For example, they present the number of uterine immune cells in fig 4, but without explaining why we should care about the uterine and what these negative results mean.

Since the conclusion that iNKT cells are protective is indirect, the authors may modify their title, for example, α-GalCer stimulation could be mentioned in the title.

6. PLOS authors have the option to publish the peer review history of their article (what does this mean?). If published, this will include your full peer review and any attached files.

Reviewer #1: No

Reviewer #2: No

---

## [Author Response · Author response to Decision Letter 0]

8 Sep 2021

Comments to the Author 

Review Comments to the Author  Please use the space provided to explain your answers to the questions above. You may also include additional comments for the author, including concerns about dual publication, research ethics, or publication ethics. (Please upload your review as an attachment if it exceeds 20,000 characters)

Reviewer #1: In this manuscript, the authors investigated a role of iNKT cells as a mediator of immune responses that could contribute to fungal clearance in a mouse model of C. albicans vaginal infection. Samples of vaginal lavage fluid were collected from estrogenized mice pre-treated with α-GalCer one day prior to vaginal inoculation and examined for fungal loads, cellular infiltrates and levels of S100A8. Leukocytes isolated from uterine and vaginal tissues were analyzed for respective cell-surface markers for identification. Results from lavage fluid indicated a reduction in vaginal fungal burden on day 7 p.i. and an increase in leukocyte migration on day 3 p.i. with a positive correlation between the presence of S100A8 and leukocyte counts. An analysis of uterine- or vaginal-associated cells by flow cytometry showed elevated numbers of CD8+ T cells and NK cells in the vagina from mice treated with α-GalCer while leukocyte profiles in the uterus were similar between the sham and α-GalCer groups. Collectively, the authors concluded that iNKT cells, presumably activated by the α-GalCer treatment, mediated a protective inflammatory response by induction of S100A8 through a mechanism involving CD8+ T cells and NK cells that ultimately resulted in reduction in vaginal C. albicans burden. Despite the rigorous flow cytometry analyses, there are several weaknesses and missing elements in the methodology, and overall, the study appears incomplete to support the conclusions. The following are major issues found in the manuscript.

Response: We appreciate the reviewer’s comment. We added several experiments, and modified the manuscript and figures according to the reviewer’s comments.

 

1. The cited references in regard to priming of iNKT cells by α-GalCer demonstrate immunosuppressive properties. As shown in (21), activation of iNKT cells by intraperitoneal α-GalCer treatment resulted in reduced phagocytic capacity and increased C. albicans tissue invasion/deaths while CD1d KO mice (NKT cells-deficient) showed higher fungal clearance/survival. α-GalCer treatment alone was indicated to have a neutropenic effect (22). The contradictory outcomes in the vaginal immune response should be addressed.

Response: We appreciate the reviewer’s comment. As the reviewer pointed out, iNKT cells activated by α-GalCer could demonstrate immunosuppressive properties. The cited articles (21, 22) showed the immunosuppressive properties against infection, however, these articles focused on the systemic infections after α-GalCer stimulation. In our vulvovaginal candidiasis model, we focused on the infection in vagina (mucosal infection), therefore, site and severity of infection was different. The previous report showed that iNKT cells activated by α-GalCer played the protective role in pulmonary S. pneumoniae infection (Kawakami et al. Eur J Immunol 2003; reference 31). In addition, iNKT cells were reported to play the protective role in HSV-2 infection (23). These differences (systemic infection or local/mucosal infection) are presumed to lead to the contradictory outcomes. We added these informations in the Discussion Part (Line 339-349).

2. As mentioned by the authors, the effect of α-GalCer stimulation on iNKT cells appears to occur as fast as 2-3 h post i.p. injection. The literature also suggests that iNKT activation is short living. A kinetic study showed that iNKT activation, measured by CD69 expression as a marker, peaked at 12 h post injection and returned to the baseline level within 72 h (21). Similar results were seen in fungal (22) and HSV2 (23) loads increased within 2 days. A study in (24) included α-GalCer injection on -2, 0, 3, 7 days post-infection and showed no difference in chlamydia burden. Thus, the effect of α-GalCer on iNKT cell activation on the time points is likely minimal and should be optimized/monitored/confirmed in the mouse VVC model.

Response: We appreciate the reviewer’s comment. Indeed, iNKT cell activation was known to be short-living, therefore, whether α-GalCer injection prior to C. albicans inoculation induce leukocyte infiltrations and reduce vaginal fungal burden is unclear. According to the reviewer’s comment, we performed additional experiments with different injection timing of α-GalCer and analyzed vaginal fungal burden 7 days after C. albicans inoculation. In detail, we intraperitoneally injected α-GalCer 1 or 3 day after vaginal C. albicans inoculation, and compared with the sham group. The results showed that there were no significant differences between α-GalCer group and the sham group, although the number of used mice was limited. In addition, we compared vaginal fungal burden among Day -1 α-GalCer injected experiment (Figure 2B data) and the above additional data. There were no significant differences among these groups, although the mean of fungal burden was seemed to be higher in Day 3 α-GalCer injected experiment. Taken together, it seemed that the effect of α-GalCer stimulation reducing vaginal fungal burden was more apparent when α-GalCer was injected earlier before or after C. albicans inoculation, therefore, we assumed that our experimental model was thought to be appropriate to assess the effect of α-GalCer stimulation and iNKT cell role in vulvovaginal candidiasis.

3. Despite the high estrogen dose and C. albicans inoculum, seemingly higher ends of the standard range, it is concerning that animals failed sustain vaginal fungal burden for a period of 14 days in conventional C57BL6 mice. Although SC5314 strain is not a robust colonizer of the vaginal mucosa, estrogenized mice should maintain consistent levels of colonization for 14 days, and longer in most cases. Or the data are not interpretable if there is no distinction between clearance and lack of colonization.

Response: We appreciate the reviewer’s comment. As mentioned, C. albicans SC5314 strain is not a robust colonizer of the vaginal mucosa, although this strain is used as the reference strain. In our study, the duration and fungal burden was seemed to be less than those in the previous studies, however, C. albicans could be detected from most of the mice in the sham group 14 days after C. albicans inoculation. In addition, our results showed that C. albicans was detected from vaginal lavage 3-days after inoculation, which meant no mice had lack of colonization (although, the fungal burden in the murine vagina differed). Based on these data, we thought that C. albicans colonized in murine vagina, although each mouse had different clearance rate. Previous reports also suggested that the differences of susceptibilities depended on the murine strain (Mycopathologia 2013, Med Mycol 2003; reference 45, 46). In addition, symptoms of vulvovaginal candidiasis vary considerably from person to person in human, therefore, our models are rather close to clinical settings and might be useful. Moreover, the differences of fungal burden might be affected by the differences of the environments of the facilities. Collectively, we presumed that C. albicans clearance occurred more frequently in the α-GalCer-stimulated mice, although the fungal burden was seemed to be less than those in the previous studies. We added and modified the manuscript according to the above discussion. (Line 389-397)

4. It is unclear whether trypan blue dye was used for the purpose of cell viability staining or pan-nuclear staining. Since trypan blue dye is only permeable in cells the lacking intact membrane integrity (i.e. dead), it is only helpful in identifying dead cells. To accurately quantify a cell infiltrate, a staining method that aids visualization of the nuclear morphology (e.g. H&E or Pap smear) should be used. On the same note, “inflammatory cells” should be reworded to “leukocyte infiltrates”.

Response: We appreciate the reviewer’s comment. First, we reworded “inflammatory cells” to “leukocyte infiltrates” as the reviewer suggested. In addition, as the reviewer mentioned, Pap smear is commonly used to quantify cell infiltrate, therefore, we performed the additional experiments about Pap smear. The Figure 2D showed the results of leukocyte infiltrates analyzed by Pap smear. We compared the number of infiltrating leukocytes between α-GalCer-stimulated group and Sham group, which showed that leukocyte infiltrates were significantly higher in α-GalCer-stimulated group than those in Sham group. In addition, we conducted the correlation analysis of leukocyte infiltrates between Pap smear and trypan blue-based count. The Figure 2E showed the significant positive relationships between these two methods both in α-GalCer-stimulated group and in Sham group. These results clearly showed that leukocyte infiltrates were significantly higher in α-GalCer-stimulated mice, and additionally it was shown that trypan blue-based counting could be used to assess leukocytes in vaginal lavages. We added and modified the Material and Methods, and Results according to the above results of the additional experiments. (Line 146-150, 220-227, 237-242)

5. Previous studies showed that infiltrating cells into the vaginal lumen following vaginal C. albicans inoculation are predominantly neutrophils [PubMed 15102806], are the main source of S100A8 detected in vaginal secretions (11) and have no apparent effect on fungal burden in mice [28292981]. This information should be addressed in Discussion. Furthermore, it is unknow whether iNKT cell priming by α-GalCer or any downstream effectors (CD8+ T cells and NK cells) have impact on the expression of S100A8. Since S100A8 is the sole parameter of antifungal activity in the current study, this should be confirmed experimentally.

Response: We appreciate the reviewer’s comment. As the reviewer mentioned, the infiltrating cells into the vaginal lumen was mostly neutrophils, and these infiltrating neutrophils secreted S100A8. Our results (Figure 2 and 3) showed that the number of infiltrating cells (neutrophils) into vagina was significantly higher in the α-GalCer-stimulated mice and S100A8 was positively correlated with the number of infiltrating cells (neutrophils) in the vaginal lavage, which implicated that α-GalCer injection led to more neutrophil infiltration, which resulted in the S100A8 secretion. However, this secreted S100A8 was reported to be associated with fungal clearance. In addition, as the reviewer mentioned, it is unknown whether iNKT cells activated by α-Galcer or CD8+ T-cells/NK cells have impact on the expression of S100A8. To assess iNKT cell activation, we evaluated IFNγ in vaginal lavage, which showed that the concentration of IFNγ in vaginal lavage was under detection limit. Similarly, the number of iNKT cells and other CD8+ T-cells/NK cells was low, therefore, it was hard to investigate effects of these cells on S100A8. Additional investigation showed that the number of iNKT cells increased after α-GalCer injection, which might be related with leukocyte infiltrations and fungal clearance, although this rigrous mechanism was not unclear. Accordingly, we assumed that S100A8 and other factors cooperatively contributed to the fungal clearance, however, exact mechanism of this clearance still remain unknown. We cited several articles and discussed these points and limitations in the Discussion Part. (Line 357-367, 397-406)

6. What is the rationale for evaluating the immune cells in the uterus in conjunction with the vaginal cells? C. albicans from the vaginal origin rarely invades the upper reproductive tract or does not lead to infection. If so, this should be reflected in the data from the uterine immune cells. If not, the data are not relevant in the current study and should be removed.

Response: We appreciate the reviewer’s comment. The previous report showed that uterine γδT-cells played the important role in the protection against C. albicans vaginal infection, therefore, we evaluated the uterine immune cells in this study. As we showed data in the Figure 4, the number of the immune cells in the uterus did not differ between α-GalCer-stimulated group and the Sham group. We did not mention these points, therefore, we added and modified several sentences in the Result and Discussion part. (Line 303-306, 369-372).

 

Reviewer #2: In the manuscript, Abe et al. studied the host defense with a focus on the effect of iNKT cell activation in a murine model of vulvovaginal candidiasis. They found that mice receiving α-GalCer, an iNKT cell agonist, control fungal pathogen better than mice receiving PBS. This better fungal clearance is accompanied by increased CD8+ T cells and NK cells. There is also a trend in increased S100A8 production but it does not reach statistical significance. The overall research design and results are solid. There are several concerns about this manuscript:

Response: We appreciate the reviewer’s comment. We added experiments, and modified the manuscript and replied to the concerns as follows.

A major concern is that it is unclear whether iNKT cells are induced or activated in their model, for example, if their cell number changes, or if they produce more cytokines after stimulation.

Response: We appreciate the reviewer’s comment. According to the reviewer’s suggestion, we evaluated the exact number of iNKT cells in murine vagina. The Figure 2B (New figure) and C showed the iNKT cell numbers and percentage in CD45+ hematopoietic cells in vagina, which indicated that the number of iNKT cells in α-GalCer-stimulated mice was significantly higher than that in Sham group. In addition, the rate of iNKT cells in CD45+ cells was also significantly higher in α-GalCer-stimulated mice. These results implicated that iNKT cells were induced by α-GalCer stimulation in our model. Moreover, we also evaluated cytokine in the vaginal lavage (IFNγ, which was the major cytokine secreted from activated iNKT cells), however, the amount of IFNγ was quite low in the vaginal lavages and below the limitation of detection both α-GalCer-stimulated group and Sham group. Collectively, we presumed that iNKT cells were induced by α-GalCer stimulation in our model. We added these data and modified the manuscript according to the above results (Line 287-293, 298-301, 318-321, 372-377, 409, Table).

Another major concern is that the author presents the data without explaining clearly the purpose or the meanings. For example, they present the number of uterine immune cells in fig 4, but without explaining why we should care about the uterine and what these negative results mean.

Response: We appreciate the reviewer’s comment, and apologize for using figure without explanation. The previous article showed that uterine γδT-cells played the important role in the protection against C. albicans vaginal infection, therefore, we evaluated the uterine immune cells in this study. As we showed data in the Figure 4, the number of the immune cells in the uterus did not differ between α-GalCer-stimulated group and the Sham group. We did not mention these points, therefore, we added and modified several sentences in the Result and Discussion part. (Line 303-306, 369-372).

.

Since the conclusion that iNKT cells are protective is indirect, the authors may modify their title, for example, α-GalCer stimulation could be mentioned in the title.

 Response: We appreciate the reviewer’s comment. We presumed that the role of activated iNKT cells was indirect in our study model, therefore, we modified the title as follows: “α-galactosylceramide-stimuated invariant natural killer T-cells play a protective role in murine vulvovaginal candidiasis by Candida albicans”

---

## [Decision Letter · Decision Letter 1]

18 Oct 2021

α-galactosylceramide-stimuated invariant natural killer T-cells play a protective role in murine vulvovaginal candidiasis by *Candida albicans*

PONE-D-21-16033R1

Dear Dr. Miyazaki,

We’re pleased to inform you that your manuscript has been judged scientifically suitable for publication and will be formally accepted for publication once it meets all outstanding technical requirements.

Kind regards,

Karen L. Wozniak, PhD

Academic Editor

PLOS ONE

Additional Editor Comments (optional):

Reviewers' comments:

Reviewer's Responses to Questions

**Comments to the Author**

1. If the authors have adequately addressed your comments raised in a previous round of review and you feel that this manuscript is now acceptable for publication, you may indicate that here to bypass the “Comments to the Author” section, enter your conflict of interest statement in the “Confidential to Editor” section, and submit your "Accept" recommendation.

Reviewer #1: All comments have been addressed

Reviewer #2: All comments have been addressed

2. Is the manuscript technically sound, and do the data support the conclusions?

Reviewer #1: Yes

Reviewer #2: Yes

3. Has the statistical analysis been performed appropriately and rigorously? 

Reviewer #1: Yes

Reviewer #2: Yes

4. Have the authors made all data underlying the findings in their manuscript fully available?

Reviewer #1: Yes

Reviewer #2: Yes

5. Is the manuscript presented in an intelligible fashion and written in standard English?

Reviewer #1: Yes

Reviewer #2: Yes

6. Review Comments to the Author

Reviewer #1: (No Response)

Reviewer #2: (No Response)

7. PLOS authors have the option to publish the peer review history of their article (what does this mean?). If published, this will include your full peer review and any attached files.

Reviewer #1: No

Reviewer #2: No

---

## [Editor Report · Acceptance letter]

5 Nov 2021

PONE-D-21-16033R1 

α-galactosylceramide-stimuated invariant natural killer T-cells play a protective role in murine vulvovaginal candidiasis by *Candida albicans*

Dear Dr. Miyazaki:

I'm pleased to inform you that your manuscript has been deemed suitable for publication in PLOS ONE. Congratulations! Your manuscript is now with our production department. 

Kind regards, 

on behalf of

Dr. Karen L. Wozniak 

Academic Editor

PLOS ONE